# Phase nucleation through confined spinodal fluctuations at crystal defects evidenced in Fe-Mn alloys

A. Kwiatkowski da Silva [1], D. Ponge[1], Z. Peng[1], G. Inden[1], Y. Lu[2], A. Breen[1], B. Gault [1] & D. Raabe[1]

Analysis and design of materials and fluids requires understanding of the fundamental relationships between structure, composition, and properties. Dislocations and grain boundaries influence microstructure evolution through the enhancement of diffusion and by facilitating heterogeneous nucleation, where atoms must overcome a potential barrier to enable the early stage of formation of a phase. Adsorption and spinodal decomposition are known precursor states to nucleation and phase transition; however, nucleation remains the less well-understood step in the complete thermodynamic sequence that shapes a microstructure. Here, we report near-atomic-scale observations of a phase transition mechanism that consists in solute adsorption to crystalline defects followed by linear and planar spinodal fluctuations in an Fe-Mn model alloy. These fluctuations provide a pathway for austenite nucleation due to the higher driving force for phase transition in the solute-rich regions. Our observations are supported by thermodynamic calculations, which predict the possibility of spinodal decomposition due to magnetic ordering.

---

[1] Max-Planck-Institut für Eisenforschung, Max-Planck-Strasse 1, 40237 Düsseldorf, Germany. [2] Database Systems and Data Mining Group, Ludwig-Maximilians-Universität München, Oettingenstraße 67, 80538 München, Germany. Correspondence and requests for materials should be addressed to A.K.d.S. (email: a.kwdasilva@mpie.de)

Spinodal decomposition and diffusional nucleation and growth are the two different mechanisms by which a homogeneous supersaturated matrix phase can decompose into a state of lower free energy consisting at the end of a heterogeneous mixture of two or more phases[1, 2]. Spinodal decomposition is one of the most important phenomena in multicomponent solids and fluids. It was first treated by Gibbs[2] who derived a necessary condition for the stability or, respectively, metastability of a phase: the solution is stable or unstable with respect to composition changes when, respectively, $\left(\partial^2 G/\partial c^2\right)_{T,P}>0$ and $\left(\partial^2 G/\partial c^2\right)_{T,P}<0$, where $\partial^2 G/\partial c^2$ is the second derivative of the free energy as function of concentration. The limit of metastability, that is, $\left(\partial^2 G/\partial c^2\right)_{T,P}= 0$, is called the spinodal. It marks the compositional maximum or minimum of the chemical potential[2–4]. Typically, solutions undergo spinodal decomposition during state changes such as those associated with heating and cooling[5]. Solutions are usually stable at higher temperatures due to the high entropy of mixing and metastable at lower temperatures. Temperature cycling of oil in water is a typical example. For the solution to decompose, the amount of solute must exceed the critical composition where $\left(\partial^2 G/\partial c^2\right)_{T,P}= 0$.

Nucleation on crystalline defects is usually described using the classical heterogeneous nucleation theory (CNT)[1, 6, 7]. CNT describes the formation of the nucleus of the new phase from solution in a single step by statistical fluctuations[1, 6, 8]. In this context, the preferential nucleation of a phase at structural imperfections such as grain boundaries and surfaces is attributed to the reduction of the total interfacial energy of the nucleus (Gibbs–Thomson effect or Laplacian pressure)[1, 7, 9]. Recently, the

tenets of CNT have been questioned by many experimental investigations on the nucleation of solids from liquid solutions (crystallization)[6, 8, 10–14]. Non-classical nucleation models introduce an additional step that precedes nucleation[6, 8]. Spinodal decomposition is the most common pathway in such multi-step nucleation sequences[6, 8].

For solid-state phase transformations, grain boundary segregation or adsorption is widely regarded as a pathway for phase transitions, since segregation to defects locally alters the thermodynamic driving force for phase transformations by changing the chemical composition of the interfacial region[15–18]. Also, many authors proposed that the interface by itself may transform in a manner similar to bulk phases, transitioning from one state to another as a function of temperature and composition[19–23]. In this context, the term "complexion" was introduced to distinguish these states from three-dimensional bulk phases[20, 21, 23–25]. Similarly, experimental and theoretical studies suggest that segregation to the free surface could lead to a solute enrichment large enough to trigger spinodal decomposition[26–28]. Only few efforts have been dedicated though to understand the role of such transitions on the nucleation of bulk phases at internal interfaces, that is, high and low-angle grain boundaries (HAGBs and LAGBs), and triple junctions, as well as at dislocations. Indeed, usually it is assumed that the spinodal decomposition mechanism cannot explain a change in the crystal structure and/or in the orientation relationship[1].

Here, we report the observation of the formation of spinodal fluctuations confined to grain boundaries and dislocations in a binary Fe-9 atomic% (at.%) Mn model alloy (body-centered cubic (BCC) martensitic solid solution) at near-atomic scale using atom

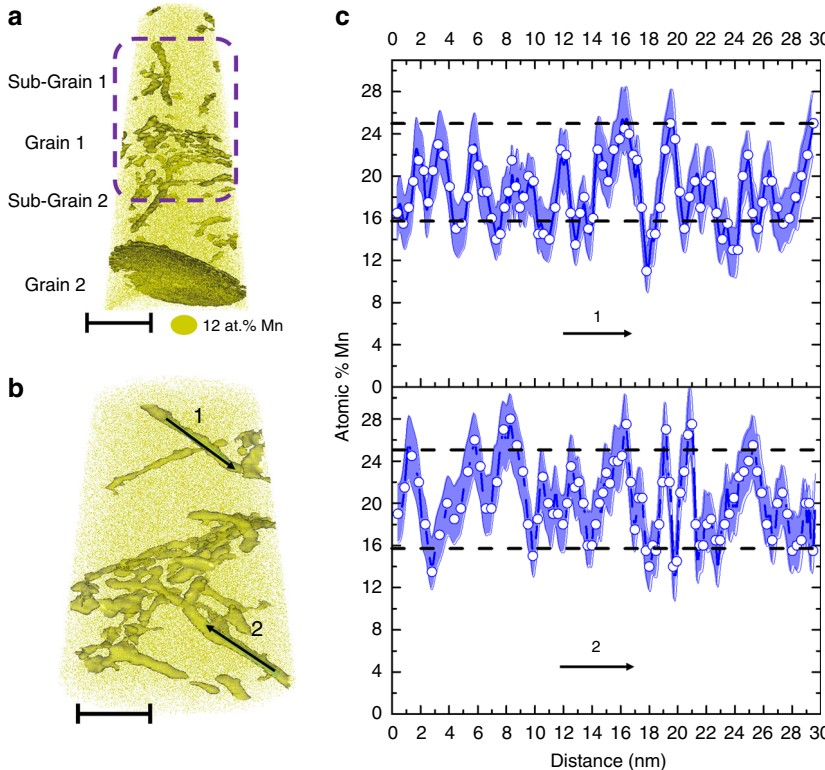

**Fig. 1** APT analysis of dislocations decorated with Mn after 6 h at 450 °C. **a** The 12.5 at.% Mn iso-concentration surfaces (12.5 atomic % Mn was chosen as a threshold value to highlight Mn-enriched regions) showing two grain boundaries and numerous dislocations decorated by Mn. Scale bar, 40 nm. **b** Close-up on the middle section of the dataset indicated by the purple dashed line showing decorated dislocations. Scale bar, 30 nm. **c** 1D composition profiles along the two dislocations marked by the arrows in **b**). The points are the experimental values, which are connected by blue lines for better visualization. The blue shading represents the statistical error associated with the calculation of the composition. The dashed lines show the typical values of compositional fluctuation

probe tomography (APT). The bulk concentration in the adjacent grain interiors is outside the region of metastability, hence the solution does not decompose inside the abutting crystals. This is different for the interfaces between them or dislocations within these grains. Mn segregates, that is, adsorbs to lattice defects such as grain boundaries and dislocations driven by the Gibbsian trend to minimize the system's free energy[29–33]. The Mn concentration on the interfaces increases as the aging proceeds at moderate temperature and, once the critical composition of metastability is reached, local fluctuations occur that tend to grow with time. This means that the linear and planar spinodal observed here are similar to the well-known bulk spinodal, with the exception that the solution's metastability is reached only within the region where Mn segregates at dislocations or at the interface but not in the adjacent bulk. These low-dimensional spinodal fluctuations on the grain boundary act as a precursor to the nucleation of a new face-centered cubic (FCC) phase (austenite) when they become strong enough in composition and wavelength.

## Results

**Segregation behavior at 450 °C.** For studying these effects, we homogenized the Fe-9 at.% Mn alloy at 1100 °C and then quenched and cold-rolled it to 50% reduction in order to increase the dislocation and interface density. The alloy was annealed at

450 °C for 6 h and subsequently analyzed by APT (see Methods section). The APT reconstruction of a typical dataset, shown in Fig. 1a, reveals structural defects decorated with solute Mn: elongated, tubular features corresponding to dislocations[34] and two grain boundaries. A close-up on some of these dislocations is shown in Fig. 1b and the two one-dimensional (1D) composition profiles plotted in Fig. 1c were computed along the direction indicated by arrows in Fig. 1b. These dislocations present a fluctuation pattern: a base level around 15 at.% Mn together with fluctuations ranging between 15 at.% Mn and 25–30 at.% Mn.

During acquisition of this dataset, the pattern formed by the successive ion impact on the detector reveals elements of the local crystallography of the sample, allowing for atom probe crystallography analysis of this particular specimen[35]. Three maps calculated for $5 \times 10^6$ ions collected during the analysis show such patterns, as visible in Fig. 2 for the three individual grains imaged. The maps shown in Fig. 2b–d were obtained from the regions marked by the blue, violet, and green boxes in Fig. 2a. The corresponding stereographic projections after identification of the main poles in the map were superimposed, and they allow for a full determination of the analyzed grain with respect to the specimen's main axis as well the calculation of the misorientation between the grains[36]. The misorientation between Grain 1 and Grain 2 is 5° and between Grain 2 and

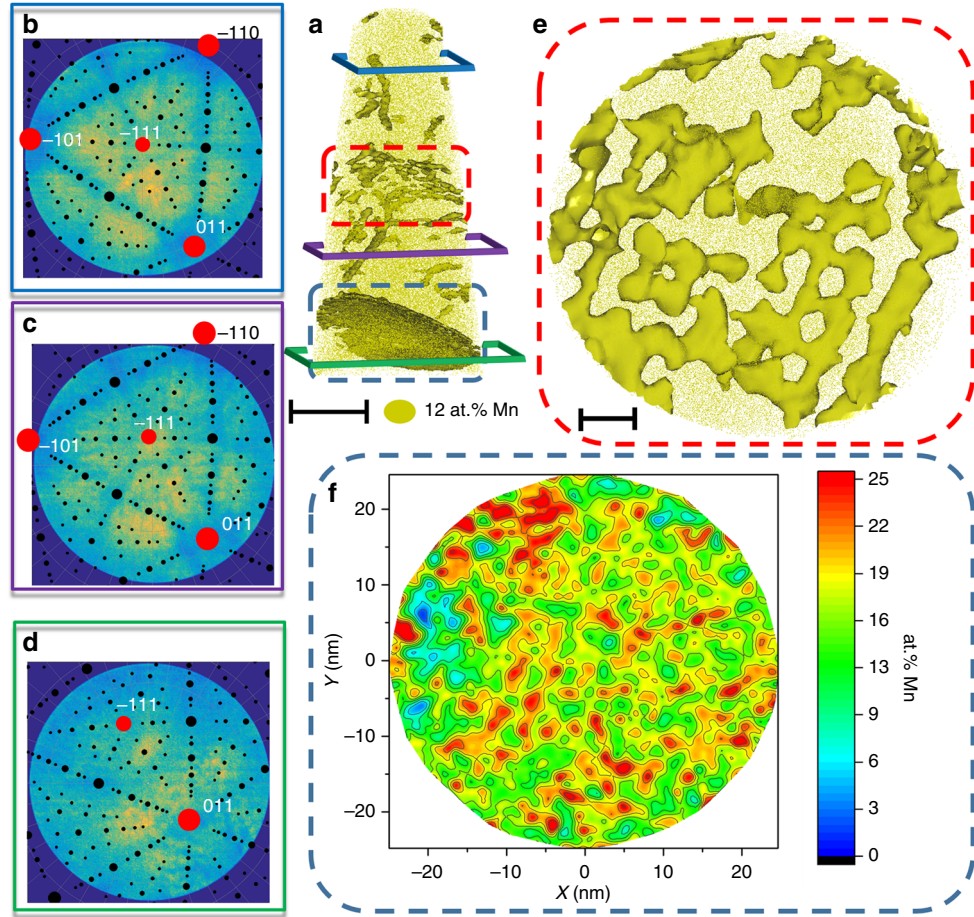

**Fig. 2** APT analysis of a grain boundary decorated with Mn after 6 h at 450 °C. **a** The 12.5 at.% Mn iso-concentration surfaces (12.5 atomic % Mn was chosen as a threshold value to highlight Mn-enriched regions). Scale bar, 40 nm. **b–d** The detector map of the regions marked by the blue, purple, and green frames in **a**, respectively. The corresponding stereographic projections (black dots) after identification of the main poles in the map (red dots) were superimposed. **e** Detail of the twisted LAGB marked by the red dashed box in **a** as revealed by the iso-concentration surfaces. Scale bar, 10 nm. The grain boundary highlighted by the blue dashed box in **a** was exported and displayed in **f** using an in-plane concentration analysis inside the grain boundary plane

Grain 3 is 11.5°. Fig. 2e shows the LAGB in plane, indicating that it is constituted by dislocations forming a chess-board array. Such dislocation pattern indicates that this is a twisted LAGB. Each of these dislocations shows compositional fluctuations similar to the individual dislocations studied in Fig. 1c. The second interface is a high-angle grain boundary (HAGB), since the dislocation spacing would be smaller than 10 Burgers vectors[37], which exhibits a completely different segregation behavior. To best describe this segregation behavior on the interface, we firstly established the grain boundary plane based on the local center of mass of the solutes. Then the local composition was derived from a cumulative curve computed along the local normal of the grain boundary plane (see Methods part for details). These results are summarized in the two-dimensional (2D) map shown in Fig. 2f. We analyzed multiple segregated regions in datasets obtained from different HAGB and we observed that all display a very similar compositional pattern of adsorption: the segregation of Mn to the interfaces assumes a base level around 15 at.% Mn together with fluctuations ranging between 15 at.% Mn and 25–30 at.% Mn. The linear and planar compositional fluctuations observed here are visually very similar to those reported for bulk spinodal decomposition[38–41] and contrast with other studies that show preferential segregation of solutes to periodically spaced sites at grain boundaries[42–45].

**Thermodynamic calculations**. For better understanding the observed segregation behavior at defects, we investigated the underlying local thermodynamics. According to the Gibbs adsorption theory[46], the driving force for segregation is the minimization of the total free energy of the system, here primarily driven by the reduction of the free energy of the defect or interface via partitioning of the solute atoms, a phenomenon referred to as equilibrium segregation or adsorption[47]. Gibbs described interfaces as having a "phase-like" behavior in order to explain the adsorption phenomena and obtained a simple expression for the free energy of the interface[25]. It was also shown that segregation to linear defects exhibits a similar behavior[29, 34]. When derived for a single temperature, this expression is known as the Gibbs adsorption isotherm: $Ad\sigma + \sum_i^n d\mu_i = 0$, where $A$ is the dividing area of the interface, $\sigma$ is the interface energy per unit area, $n_i$ is the number atoms of a given element, and $\mu_i$ is the chemical potential of a given element in the boundary[30]. The Gibbs adsorption isotherm potentially allows the quantitative description of segregation to defects, yet its applicability is limited as the defect or interface energy is often not known as a function of temperature and concentration[48]. Likewise, the actual potential difference between the bulk and the defect is both of a mechanical and chemical nature owing to the Gibbs–Thomson capillarity effect and internal elastic misfit stresses. Irrespective of these more complex conditions, we consider here at first the chemical

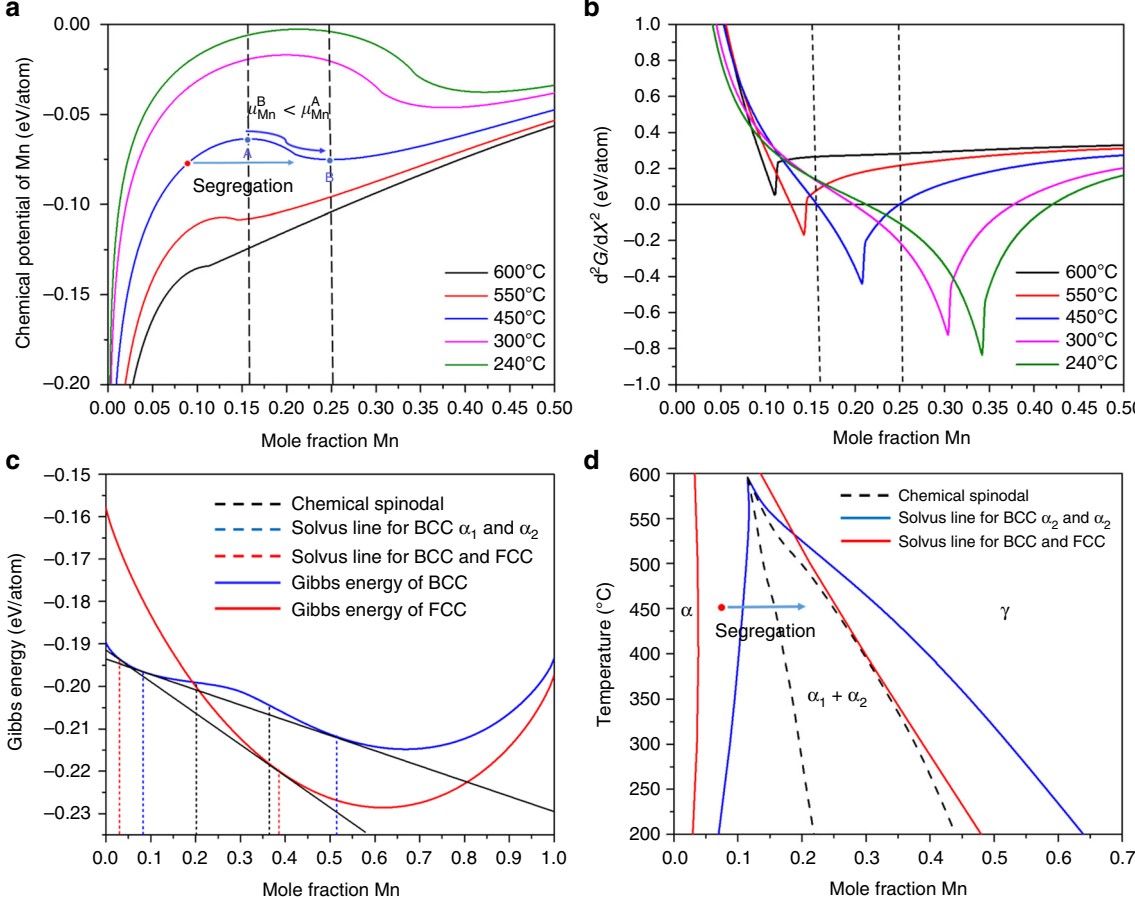

**Fig. 3** Thermodynamic calculations for the BCC Fe-Mn bulk system. **a** Chemical potential of Mn as a function of the mole fraction of Mn for different temperatures. The black dashed lines indicate the spinodal points at 450 °C. **b** Second derivative of the Gibbs free energy in function of the molar fraction of Mn. **c** Gibbs free energy of the BCC and FCC phases at 300 °C showing the bulk chemical spinodal of the BCC phase and the local equilibrium obtained by the tie-lines among the different phases. **d** Fe-Mn bulk phase diagram including the chemical spinodal, the solvus lines between BCC and two spinodal compositions $\alpha_1$ and $\alpha_2$ and the solvus lines for the composition in equilibrium between BCC (ferrite) and γ FCC (austenite). The red point in **a**, **d** indicates the initial composition of the system and the arrows indicate the solute enrichment due to segregation

potential of Mn in the bulk in order to best describe the segregation behavior.

We calculated the chemical potential of Mn in the BCC bulk matrix as a function of the mole fraction of Mn for different temperatures (between 600 °C and 240 °C) using the Thermocalc TCFE7 thermodynamic database for iron-based solid solutions (Fig. 3a). For this range of temperatures and compositions, the chemical potential is always smaller than zero; therefore, segregation to defects is likely since it implies the minimization of their free energy. Fig. 3a shows that the chemical potential reaches a maximum and then tends to decrease as a function of the Mn concentration, indicating the presence of a chemical spinodal in this range of temperatures for the BCC bulk phase. The chemical spinodal is the second derivative of the Gibbs free energy (Fig. 3b). The data show that the metastable region, that is, the region inside the chemical spinodal, extends at 450 °C between 15.8 and 25 at.% Mn. This range is in very good agreement with the compositional fluctuations measured at the imperfections for this temperature, as reported in Figs. 1 and 2. Assuming the existence of a miscibility gap only for the BCC phase, we calculated an Fe-Mn BCC phase diagram from the tie-lines between the Mn poor ($\alpha_1$) and rich ($\alpha_2$) regions (Fig. 2c). Fig. 3d shows an Fe-Mn phase diagram that superimposes the solvus lines for the BCC $\alpha_1$ and $\alpha_2$ regions, the chemical spinodal for the BCC phase and the solvus lines for the equilibrium between the BCC phase and the FCC (austenite) phase in equilibrium. The distortion of the miscibility gap (assuming a horn-like shape) is typical for systems with magnetic ordering contributions[49, 50].

The thermodynamic data in Fig. 3a–d allow to distinguish between two regimes of interface segregation. For $\left(\partial^2 G/\partial X^2\right) > 0$ regular grain boundary segregation proceeds in a compositionally uniform way by enriching solute Mn that stems from the adjacent grains at the interface. For $\left(\partial^2 G/\partial X^2\right) < 0$ the segregation proceeds by forming local fluctuations, since the compositions inside the chemical spinodal are metastable. These two cases concern the chemistry of the segregation. Additional heterogeneity affecting compositional variations can arise from structural features of the interface such as dislocations, curvature, or kinks.

**Segregation behavior at 240 °C.** In order to further check the hypothesis of a confined spinodal, we conducted additional heat treatments to study the effect of the chemical spinodal on the resulting segregation patterns. We annealed the Fe-9 at.% Mn alloy for 6 h at 450 °C and slowly cooled (<1 °C/min) it down to 240 °C and kept the sample at this temperature for 24 h. Since the compositions inside the chemical spinodal are energetically metastable, we expected that, after this thermal cycle, the concentration of the solute Mn would, at least for some of the boundaries, follow the lines of concentration calculated and shown in Fig. 3. Figure 4 indeed exemplifies such a case. The grain boundary region highlighted in Fig. 4a is shown in Fig. 4b in cross-section. The composition of this grain boundary is shown in Fig. 4c in cross section and in Fig. 4d in plane. The misorientation of this section of the boundary was estimated to be 12°, and the local composition was extracted according to the same method as in Fig. 2. We observed that the concentration clearly fluctuates between the two spinodal points (20 at.% Mn and 40 at.% Mn) shown in Fig. 3b. Further, we observe that the concentration features tend to be geometrically aligned along one preferred direction, forming linear features. We assume here that the strain field can modulate the compositional fluctuation morphology as proposed for spinodal decomposition in thin films[51].

**2D simulations.** Following these thermodynamic considerations and experimental observations we performed corresponding 2D simulations of a spinodal decomposition of an Fe-20 at.%Mn alloy at 450 °C for comparison with the segregation pattern observed experimentally at grain boundaries. We simulated the problem by solving the continuous Cahn–Hilliard equation[4, 52] using the time marching scheme developed by Eyre[53, 54] and a third-order polynomial function to represent the chemical potential of Mn as a function of the Mn content (Fig. 3a). The simulation results are shown in Fig. 5a, c. Fig. 5a is a concentration map obtained after 300 s of simulation time. Fig. 5b is an experimental concentration map obtained by APT for a grain boundary from the same condition as the data shown in Fig. 1. Fig. 5c shows composition profiles from the cross-section of the 2D map for different simulation times. Fig. 5d shows a composition profile from the cross-section of the experimental map

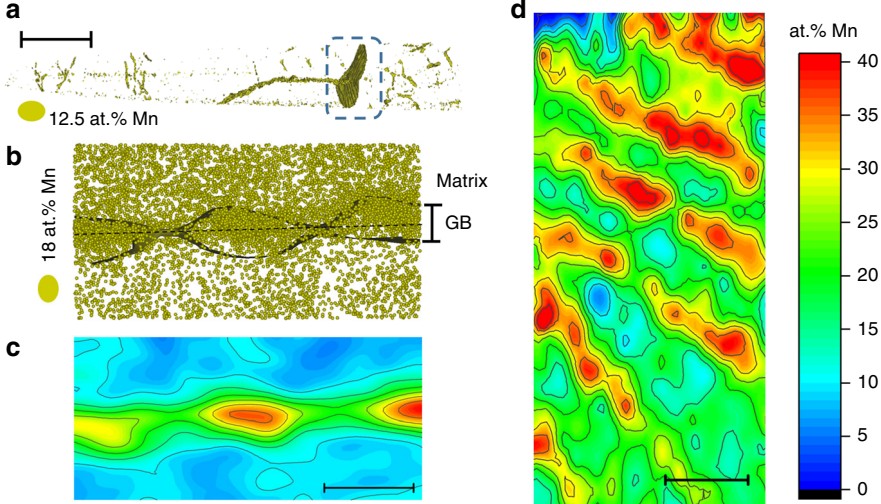

**Fig. 4** APT analysis of a grain boundary decorated with Mn after 24 h at 240 °C. Atom probe tomography results: in **a** atom probe reconstruction and 12.5 at.% Mn iso-concentration surfaces. Scale bar, 80 nm. **b** A cross-section of the grain boundary. Scale bar, 2 nm. Every yellow sphere represents a Mn atom. **c** Chemical composition of the cross-section shown in **b**. Scale bar, 5 nm (**d**): an in plane concentration analysis of the grain boundary. Scale bar, 10 nm. The analysis in **b**–**d** refer to the grain boundary highlighted in **a** using a dashed blue box

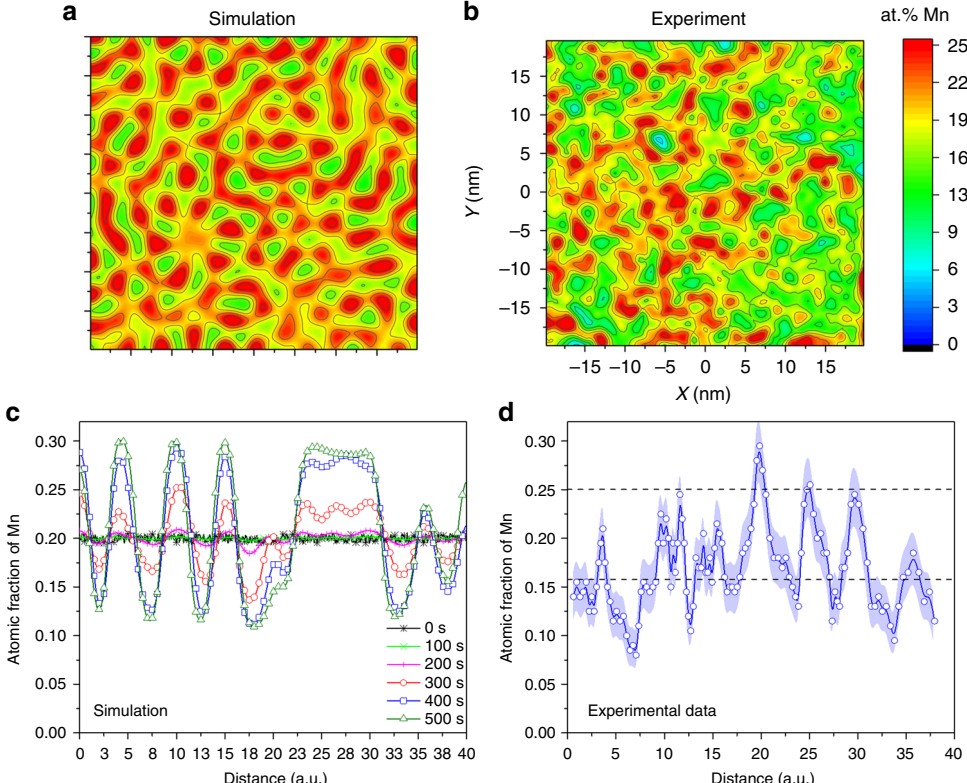

**Fig. 5** Comparison between simulation and experiments. **a**, **c** 2D simulation of a spinodal decomposition using the continuous Cahn–Hilliard equation. **b**, **d** Experimental results (APT) for a grain boundary from Fe-9 atomic % Mn solid solution, 50% cold-rolled, and annealed at 450 °C for 6 h. The **a**, **b** maps have the same color bar. The map in **a** is a concentration map for 300 s of simulation time. The **c**, **d** concentration profiles were obtained from the cross-section of **a** and **b**, respectively. **c** composition profiles for different simulation times. The points in **d** are the experimental values, which are connected by blue lines. The blue shading represents the statistical error associated with the calculation of the composition. The black dashed lines show the values of the chemical spinodal

shown in Fig. 5b. Comparison of the two figures shows that the experimental values resemble the simulated ones especially at intermediate simulation times. Nevertheless, it is important to acknowledge here that this modeling does not represent the very same situation as the segregation induced spinodal fluctuation observed experimentally, since the spinodal fluctuations evolve during the segregation and not from a homogenous Fe-20 at.% Mn composition.

**Electron microscopy and diffraction analysis**. We performed crystallographic measurements using transmission Kikuchi diffraction and electron back-scattered diffraction in order to characterize the temporal evolution of the distribution of the phases at 450 °C. Fig. 6a shows a phase map plus an image quality (IQ) map in grayscale. Fig. 6b shows an inverse pole figure of the normal direction of the same region shown in Fig. 6a. We clearly detected a small volume fraction of γ austenite (FCC phase) growing along some of the HAGBs. It is expected that austenite would start to nucleate at HAGBs and triple junctions, since these interfaces have higher driving force for segregation and the fluctuations would evolve first to a critical nucleus. Such result could be potentially explained using CNT as well, since the HAGB would have a higher contribution to the reduction of the interface energy of the nucleus. In order to fully characterize the different nucleation sites of austenite, we performed measurements using high-resolution EBSD (HR-EBSD) in a sample annealed for 2 months at 450 °C. Fig. 6c shows a phase map plus

an IQ map in grayscale. Fig. 6c shows as well the grain boundaries classified in LAGB (5°–15°) and HAGB (>15°). Fig. 6d shows an inverse pole figure of the normal direction of the same region shown in Fig. 6c. We clearly detected a high number of γ austenite grains (FCC phase) growing along both the HAGBs and LAGBs. These results suggest that the LAGB, like the ones displayed in Fig. 1, can as well act as nucleation sites for austenite.

**Composition of the austenite nucleus**. The experimental characterization of the exact event of nucleation is challenging and has not been performed so far for any material. Nevertheless, we identified by APT a few cases that resemble a recently formed nucleus of austenite at the initial stage of growth. Fig. 7a illustrates such a case. Fig. 7b presents a 1D compositional profile along the cylinder showed in Fig. 7a crossing the apparently recently formed nucleus of a different phase. This austenite nucleus seats on a LAGB or a triple junction of LAGB. Figure 7c shows a similar example, but for a sample annealed 2 weeks at 450 °C. Fig. 7d shows a 1D compositional profile along the cylinder showed in Fig. 7c crossing the austenite nucleus apparently in the initial stage of growth. In both cases, the nucleus composition is around 32 at.% of Mn. This value is very close to the expected equilibrium composition of the Mn-rich phase (32.3 at.% of Mn) obtained by spinodal decomposition (according to the phase diagram in Fig. 3d). Such results support the hypothesis that the austenite is nucleated from the Mn-rich spinodal fluctuations driven by segregation.

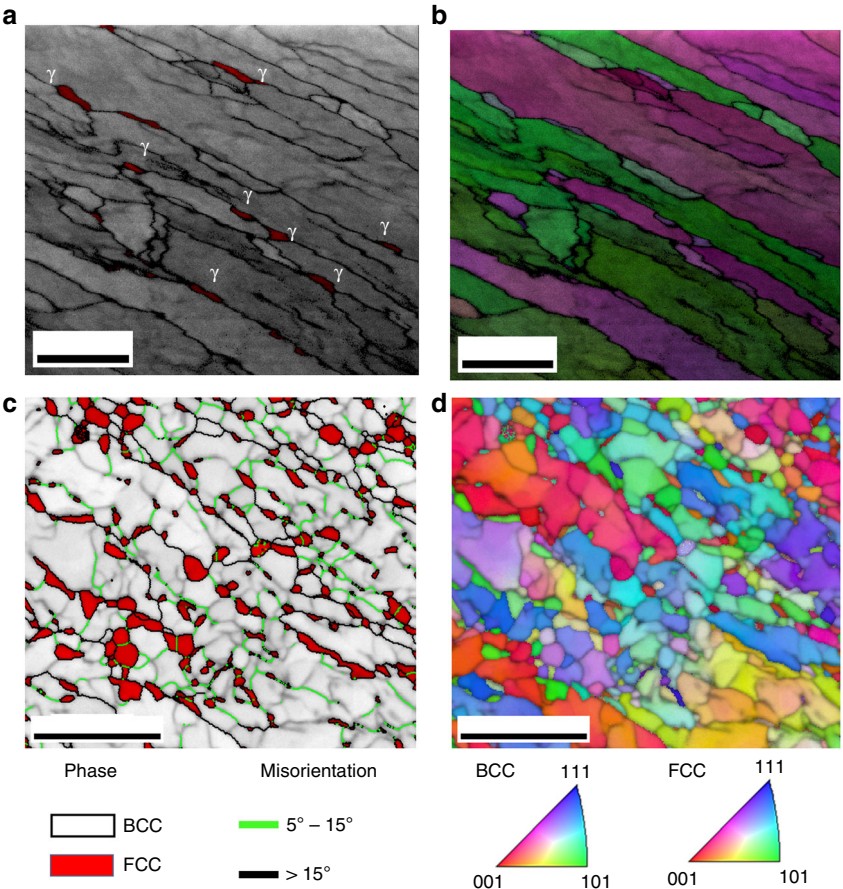

**Fig. 6** Diffraction analysis of the alloy after 6 h and 2 months at 450 °C. **a**, **b** TKD analysis of the Fe-9 Mn alloy after 6 h at 450 °C. **a** Phase map plus image quality map in grayscale and **b** inverse pole figure of the normal direction of the same region shown in **a**. Scale bar, 500 nm. **c**, **d** EBSD analysis of the Fe-9 Mn alloy after 2 months at 450 °C. **c** Phase map plus an image quality map in grayscale and **d** inverse pole figure of the normal direction of the same region shown in **c**. Scale bar, 5 μm. Austenite clearly nucleates in both high-angle and low-angle grain boundaries. γ represents the FCC-austenite phase

**Composition of austenite in near-equilibrium condition.** In order to characterize the austenite composition in near-equilibrium condition, we performed APT measurements in an Fe-9 at.% Mn sample annealed for 2 months at 450 °C. Fig. 8a shows an APT reconstruction for this stage of annealing. The APT reconstruction and a 2D compositional map show an austenite grain in stage of growth in local equilibrium with the surrounding BCC matrix. Fig. 8b provides a 1D compositional profile along the cylinder crossing the FCC–BCC interface presented in Fig. 8a. The composition of the austenite is close to the equilibrium composition for this phase (25.5 at.% Mn). The composition of the depleted zone in the BCC region is close to the equilibrium composition for the BCC phase (3.8 at.% Mn). This observation indicates that the movement of the interface is controlled by the local equilibrium at the interface. No dislocation, HAGB or LAGB decorated by solute Mn were observed for this annealing condition. These results suggest that the segregation and the spinodal fluctuations are transient states that can act as a pathway for phase transition, but they are not by themselves part of the equilibrium state of the system.

To quantitatively evaluate the impact of the spinodal fluctuations on the nucleation event, we calculate the driving force for austenite formation from an Fe-9 at.% Mn BCC matrix. The results are shown in Fig. 8c. The partitioning driving force is the difference in the free energy between the Fe-9 at.% Mn BCC matrix and the free energy of the equilibrium point of the mixture of ferrite and austenite. The partitioning driving force describes how far the system is away from equilibrium, but it is not a good

measure of the free energy available for nucleation. The partitionless driving force is the difference in free energy between an Fe-9 at.% Mn BCC phase and the austenite phase with the same composition. The partitionless driving force is typically used to evaluate the possibility of massive or martensitic transformations. The spinodal-assisted driving force is the difference in free energy between the Mn-rich BCC phase and the austenite phase with the same composition. The results show that a much higher amount of free energy is available by the proposed mechanism, that is, nucleation from the Mn-rich fluctuation in the segregation zone.

## Discussion

In summary, we conducted a joint experimental and theoretical study which reveals the existence of 1D and 2D spinodal fluctuations driven by segregation and confined to internal structural defects, that is, dislocations, HAGBs and LAGBs. The phenomena observed agree with corresponding thermodynamic equilibrium and kinetic simulations of the Fe-Mn system. The discovery that segregation to interfaces can proceed in a spinodal-like fashion for specific systems represents a fundamental contribution not only to the field of interface segregation but also to nucleation theory, since these fluctuations act as precursors to the formation of a new phase. Due to the transient nature of these confined spinodal fluctuations, they may not be regarded as equilibrium complexion states[21, 23, 24], that is, equilibrium interface states, although equilibrium interface

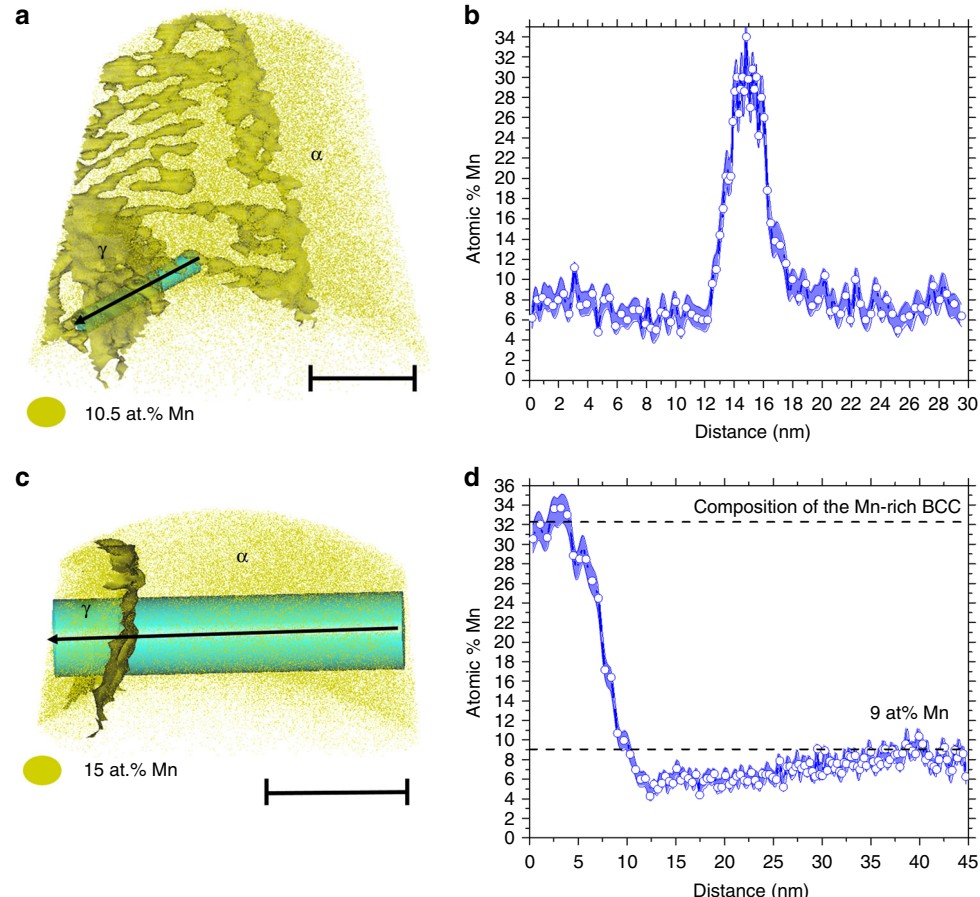

**Fig. 7** Composition of the austenite after 6 h and 2 weeks at 450 °C. **a**, **b** Fe-9 atomic % Mn solid solution, 50% cold-rolled, and annealed at 450 °C for 6 h: **a** APT reconstruction showing an austenite nucleus sited on an LAGB or a triple junction of LAGB. **b** 1D compositional profile along the cylinder showed in **a**. **c**, **d** Fe-9 atomic % Mn solid solution, 50% cold-rolled, and annealed at 450 °C for 2 weeks: **c** APT reconstruction showing an austenite nucleus in initial stage of growth. **d** 1D compositional profile along the cylinder showed in **c**. α represents the BCC-ferrite phase and γ represents the FCC-austenite phase. The points in **b**, **d** are the experimental values, which are connected by blue lines. The blue shading represents the statistical error associated with the calculation of the composition

states could possibly arise from spinodal fluctuations when the nucleation of a second phase is not possible. Most of the thermodynamic systems with positive enthalpy of mixing in the solid state and a miscibility gap at lower temperatures could follow a similar phase nucleation pathway. Examples include some of the most important metallic alloy systems for industrial, infrastructure, and mobility applications such as steels, aluminium, titanium, and magnesium alloys. Additionally, many more material systems might potentially contain such confined spinodal regions at low temperatures which may have just been overlooked so far due to limitations in assessing thermodynamic properties at low temperatures and in experimentally resolving such confined chemical–structural phenomena. In particular, in the Fe-Mn system, these fluctuations lead to the formation of austenite (FCC) observed at later stages, since a locally higher driving force becomes available through these fluctuations.

## Methods

**Sample preparation**. An ingot of the Fe-9 wt.% Mn solid solution alloy used in this work was first synthesized in a vacuum induction furnace and cast as a 4 kg rectangular billet using high-purity metals (segregated edges of the slab were cut off), annealed at 1100 °C to homogenize the primary dendritic segregation in the solidification microstructure, and then water quenched. The alloy was cold-rolled to 50% thickness reduction for increasing the dislocation and interface density. Additional long-time annealing promoting Mn diffusion and segregation was performed at 450 °C for 6 h, 2 weeks, and 2 months. A second sample was first

annealed at 450 °C for 6 h, slowly cooled down (<1 °C/min) to 240 °C, and kept at this temperature for 24 h. The nominal bulk composition was measured by wet-chemical analysis (Table 1).

**APT analysis**. APT specimens with end radii below 100 nm and the TEM lamella for TKD were prepared using an FEI Helios NanoLab600i dual-beam Focused Ion Beam/Scanning Electron Microscopy instrument. APT was performed using a LEAP 5000 XS device by Cameca Scientific Instruments, with approx. 80% detection efficiency, at a set-point temperature of 50 K in laser-pulsing mode at a wavelength of 355 nm, 500 kHz pulse repetition rate, and 30 pJ pulse energy. For reconstructing 3D atom maps, visualization, and quantification of segregation, the commercial software IVAS® by Cameca was employed following the protocol introduced by Geiser et al.[55] and detailed in Gault et al[56]. Most of the datasets were reconstructed in Voltage mode (with the exception of Fig. 7). The typical detector maps shown in Fig. 2 are strongly indicative of the high reliability of the measurement. The detector reveals elements of the local crystallography of the sample, allowing for atom probe crystallography analysis of the dataset. We could as well calibrate the reconstruction by the interplanar distance of the crystallographic planes associated with the low-hit density poles. The interface composition plots were computed using a method developed in-house in MATLAB (Mathworks Inc.). First, a DBSCAN algorithm or a nearest-neighbor filter was used to select the solute atoms that are segregated along the interface[57, 58]. According to their local density maximum (center of mass), the interface plane was defined[59]. By splitting the dataset using a Voronoi tessellation, local cumulative curves can be derived[60], and therefrom the segregation width and compositional values can be extracted.

The misorientation of the grain boundaries is estimated following the methodology described in Breen et al[36]. A stereographic projection is first manually adjusted to the detector map, and therefrom the orientation of the imaged crystal is determined with respect to the specimen's main axis. The orientation of each grain is determined, as well as the corresponding Bunge–Euler angles that are

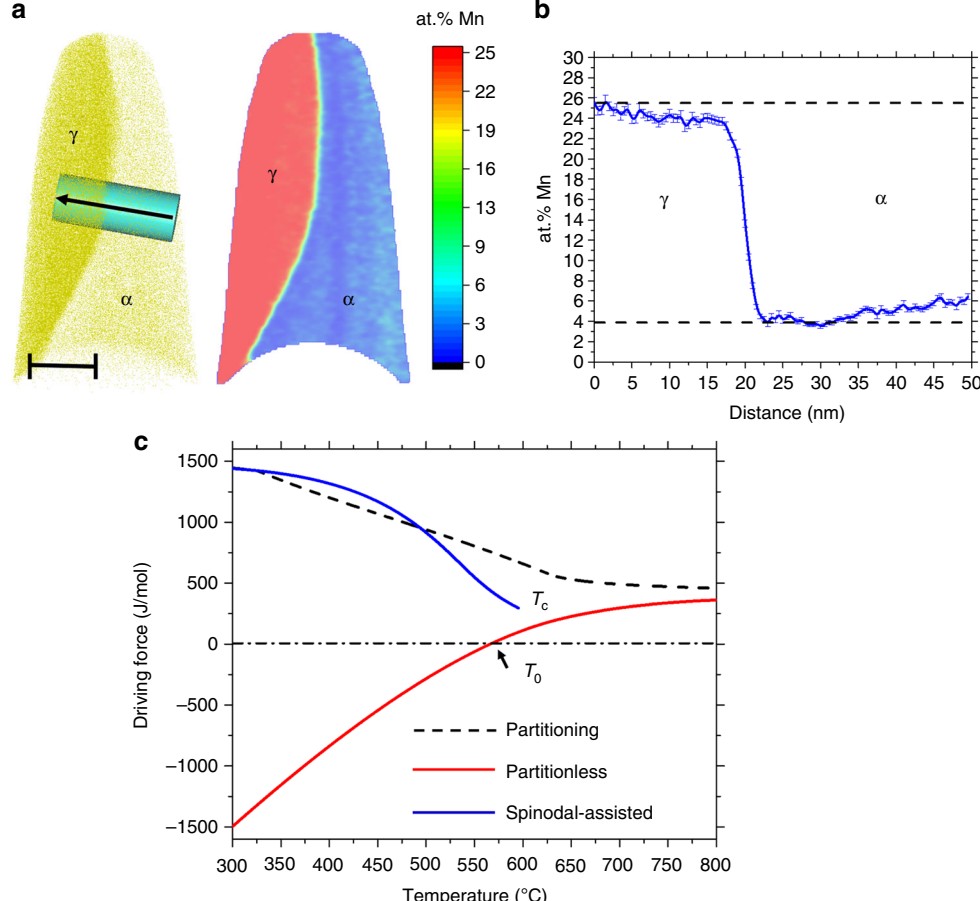

**Fig. 8** APT results after 2 months of annealing at 450 °C. **a** APT reconstruction and a 2D compositional map showing an austenite grain in stage of growth. **b** 1D compositional profile along the cylinder showed in **a**. **c** Different driving forces for austenite formation from an Fe-9 at.% Mn BCC matrix. The partitioning driving force is the difference in free energy between the Fe-9 at.% Mn BCC matrix and the free energy of the equilibrium point of the mixture of ferrite and austenite. The partitionless driving force is the difference in the free energy between an Fe-9 at.% Mn BCC phase and the austenite phase with the same composition. The spinodal-assisted driving force is the difference in free energy between the Mn-rich BCC phase and the austenite phase with the same composition. $T_c$ is the critical temperature of the miscibility gap. $T_0$ is the temperature at which the free energy of the Fe-9 at.%Mn BCC phase is equal to the free energy of the austenite phase with the same composition. $T_c$ is the critical temperature of the miscibility gap and the end of the spinodal-assisted curve. α represents the BCC-ferrite phase and γ represents the FCC-austenite phase

| Table 1 Chemical composition of Fe-9 wt.% Mn in wt.% according to wet-chemical analysis | | | | | | | | | | | |
|---|---|---|---|---|---|---|---|---|---|---|---|
| **Mn** | **C** | **Ni** | **Co** | **Mo** | **Si** | **Al** | **S** | **P** | **O** | **N** | **Fe** |
| 8.46 | 0.0075 | 0.0175 | 0.0022 | <0.002 | 0.0024 | <0.002 | 0.0047 | <0.002 | 0.0102 | 0.0040 | Balance |

subsequently used to determine the misorientation. Typical detector maps and stereograms are displayed in Fig. 2a.

**Diffraction analysis**. TKD measurements were performed using a Zeiss Merlin scanning electron microscope with a Gemini-type field emission gun electron column and Bruker's QUANTAX EBSD analysis system. The TKD measurements were performed at a step size of 2 nm and analyzed using the TSL-OIM software. HR-EBSD measurements were performed at a step size of 50 nm using a JEOL JSM-6500F field emission gun scanning electron microscope equipped with a high-speed CCD camera for pattern recording, and analyzed using the TSL-OIM software.

**Thermodynamics calculations and simulations**. The thermodynamics variables (Gibbs free energy and chemical potential of Mn) of the BCC solid solution in function of the Mn content and the phase diagrams were calculated using the Thermocalc software together with the TCFE7 thermodynamic database for iron solid solutions. The Gibbs free energy of the BCC phase is described by a Redlich–Kister–Muggianu excess model plus a function for the contribution from

magnetic ordering based on the Inden–Hillert–Jarl formalism[50, 61–64]. The coherence strain energy ($\Delta G_s$) associated with the formation of Mn-rich regions fluctuations ($\Delta X$) was evaluated according the equation[65]:

$$\Delta G_s = \eta^2 (\Delta X)^2 \frac{E}{1-\nu} V_m, \tag{1}$$

where

$$\eta = \frac{1}{a} \left( \frac{da}{dX} \right), \tag{2}$$

where $\eta$ is the fractional change in lattice parameter $a$ per unit composition change, $E$ is the Young's modulus, $\nu$ is Poisson's ratio, and $V_m$ is the molar volume of the BCC phase in function of the composition. These parameters were estimated from the values of molar volume obtained from the TCFE7 thermodynamic database and data available in the literature for the elastic constants[66, 67]. The coherence strain energy was estimated to be smaller than 0.1 eV/atom.

The 2D phase field simulations of the spinodal decomposition were computed using MATLAB (Mathworks Inc.) by solving the continuous Cahn–Hilliard

equation (3)[4, 52]:

$$\frac{\partial X}{\partial t} = D\nabla^2 \left( \frac{\partial G}{\partial X} - K^2 \nabla^2 X \right), \tag{3}$$

where $K$ is the gradient energy coefficient, $D$ is the diffusion coefficient, and $t$ is time. The gradient energy coefficient $K$ is considered as a function of the interatomic distance $\lambda$ and the regular solution parameter $\Omega$[68]:

$$K^2 = \Omega\lambda^2/2. \tag{4}$$

Since the enthalpy of the BCC magnetic solid solution cannot be described using only the regular solution model, we derived a compositional dependent $\Omega$ parameter in order to assess the value of $K$. The gradient energy coefficient $K^2$ was estimated to be of the order of $10^{-4}$ eV/nm$^2$. We used the time marching scheme developed by Eyre[53, 54] and a third-order polynomial function to represent the chemical potential of Mn as a function of the Mn content. The simulation was performed in a $400 \times 400$ cell, using a $1.73 \times 10^{-4}$ time marching parameter and a gradient energy coefficient $K$ of 0.005. The elastic energy penalty associated with the formation of Mn-rich zones was included in the calculations.

**Code availability**. The numerical code used in this study is available from the corresponding author upon request.

**Data availability**. The data supporting the findings of this study are available from the corresponding author upon request.

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

## Acknowledgements

A.K.d.S. is grateful to the Brazilian National Research Council (Conselho Nacional de Pesquisas, CNPQ) for the PhD scholarship through the "Science without Borders" Project (203077/2014-8). U. Tezins and A. Sturm are acknowledged for their support to the users of the atom probe facility at MPIE.

## Author contributions

A.K.d.S. was the lead scientist of the study; A.K.d.S., D.P., and D.R. designed the research; Z.P., B.G., and Y.L. developed the method to calculate the composition of the interfaces; A.B. and B.G. performed the APT crystallographic analysis; G.I. gave support on the thermodynamics interpretation; and A.K.d.S., D.R., and B.G. wrote the paper. All authors discussed the results and commented on the manuscript.

## Additional information

**Competing interests:** The authors declare no competing interests.

