## [Peer Review File(PDF 461 kb) · Nature Communications]

Reviewers' comments:

Reviewer #1 (Remarks to the Author):

This paper reports the discovery of a spinodal fluctuation confined to the planar interface (i.e. within the grain boundary region) in a Fe-Mn alloy. This is, to the best of my knowledge, the first such observation of this type of a grain boundary feature and it represents a significant advance to the field and therefore is fully appropriate for consideration for publication in Nature Communications. The experimental observations obtained from atom probe tomography are well supported by complementary theoretical calculations and simulations. I have a few comments for the authors to consider:

1. The opening statement on line 28 that asserts that spinodal decomposition is the most important phenomena in multi-phase materials is an overstatement in my opinion. There are lots of other phenomena that are arguably equally or more important so I would advise toning down that remark.
2. Line 44 refers to the grain boundary as "interfaces" – plural, when it should be singular "interface" since, as the authors know a grain boundary is one interface thermodynamically.
3. Line 62 refers to the grain boundaries as "high angle grain boundaries" – can they be more specific (range of angles, sigma values?).
4. A key point of the paper is that the spinodal fluctuation is confined to a planar region inside a grain boundary – can they state more precisely how the grain boundary width is defined to further validate that statement – both structurally and chemically.
5. The authors seemed to have intentionally avoided the use of the term "complexion", which is somewhat puzzling since it appears to fit perfectly with the definition of a grain boundary complexion as an equilibrium state of a grain boundary. Using the term complexion would emphasize the phase-like nature of this grain boundary spinodal. I note that this group did adopt the term complexion in a recent Science paper as applied to dislocations, so I think by not applying it consistently here it may give the wrong impression that the structure is not an equilibrium feature of the grain boundary, diminishing the novelty of the observation. I would recommend going one step further and including the term complexion in the title. This is a neat discovery of a new type of grain boundary complexion worth highlighting.
6. I am not an expert in the quantification of atom probe tomography data but I do know that reconstruction of atom positions is problematic. The changes in Mn concentration are clearly very large and not aberrations of the APT technique, but perhaps some more details about the reconstruction method (other than "we used the IVAS method") and perhaps a mass spectrum would be useful.
7. Congratulations on a nice piece of work!

Martin P. Harmer

Reviewer #2 (Remarks to the Author):

The authors report on a quasi-2D spinodal decomposition that takes place in grain boundaries in Fe-9 at% Mn alloys after a suitable heat treatment. This decomposition is observed and analyzed using 3D atom probe, which allows to map the local Mn composition in grain boundaries. These observations are discussed using thermodynamic data available for this system. These authors conclude that Mn segregation at a grain boundary increases the local Mn composition to the point where this solid solution becomes unstable and undergoes phase separation by spinodal decomposition. Some 2D simulations are presented, supporting the possibility of spinodal decomposition in 2D for this alloy system.

The results presented in the manuscript are certainly interesting and worth publishing but in my opinion they do not meet the criteria for publication in Nature Communications for several reasons.

First, the possibility that segregation at defects could lead to solute enrichment large enough to trigger spinodal decomposition is not a new finding. This pathway has been studied both experimentally and by computer simulations in the case where the defect is a free surface, see for instance C. Helms, *Surface Science* 69 (1977) 689; Y. Liu and P. Wynblatt, *Surface Science* 310 (1994) 27; C. Geng, *Surface Science* 355 (1996) 229. Furthermore the analysis provided in the paper is very limited and omits some key points, regarding for instance the role of in-plane versus out-of-plane diffusivity, whether the thermodynamic functions would correctly predict decomposition in 3D, and whether it is legitimate to use the same expressions for assessing the 2D stability of concentrated solutions in a grain boundary.

Second, the phase field modeling does not provide any new information. The authors would have to justify their choice for the atomic mobility, the interface gradient energy and other parameters if they were trying to extract from the simulations more than just a qualitative agreement, which was expected even before running the simulations.

The experimental findings themselves appear to be solid, the data well analyzed, but somewhat limited. For instance, it would have been useful to have some structural characterization of the grain boundaries, for instance by TEM, to determine whether the regions enriched in Mn correspond to grain boundary dislocation intersections or other specific features of the GBs. In addition, it would have been important to image grain boundaries at an early stage, to see whether segregation first produces a uniformly enriched Mn layer, before reaching a composition that triggered 2D decomposition. This information, it seems, would be essential to complement the present findings.

Lastly, even if all the analysis given in the manuscript is correct, it seems that this particular reaction would not be common, as the conditions required for it to proceed seem quite restrictive. The authors for instance did not discuss when this particular kinetic pathway evolved here would be overridden by the direct nucleation of precipitates in the grain boundaries, or by interface-directed spinodal decomposition (by analogy to surface-directed spinodal decomposition).

Reviewer #3 (Remarks to the Author):

Planar spinodal fluctuation at grain boundaries were studied in this manuscript. Planar spinodal fluctuation is an interesting and novel phenomenon or mechanism for grain boundary segregation. However, the manuscript is not well written, and there are many problems with it, which are detailed as follows. Consequently, the reviewer recommends rejection of the manuscript.

1) The English of the manuscript needs to be improved. There are many grammatical errors and inaccurate or confusing phrases/expressions in the manuscript.

2) The title includes the wording of "a new approach to interface manipulation", which is inappropriate. Again, an interesting phenomenon or mechanism is found for grain boundary segregation by the authors. However, this is not really "interface manipulation". Interface manipulation was not really discussed in the manuscript.

3) The abstract is not well written. The first couple of sentences are not really relevant to the study describing in the manuscript (e.g., mechanical properties or strengthening mechanisms). It is claimed that a new phase transition mechanism is discovered, however, the manuscript or the study is really on grain boundary segregation, and phase transition was barely discussed in the manuscript.

4) The introduction part of the manuscript is not well written. Essentially, there is only one paragraph for the introduction section, only describing spinodal decomposition. Background and

motivation for this study should have been clearly elucidated in the introduction section. For example, what is the current state-of-art on studies on grain boundary segregation, why is this important, and why did the authors choose Fe-Mn system.

5) In the manuscript, adsorption and grain boundary segregation are used interchangeably. Although they have some similarity, adsorption and grain boundary segregation are different. Adsorption describes the adhesion of atoms from a gas or liquid phase onto a surface, whereas grain boundary segregation describes the segregation of atoms in a solid to a grain boundary.

6) A grain boundary may not be just one plane. In the manuscript, the authors assumed that a grain boundary is just one flat plane. However, a grain boundary may be curved, and it may have multiple planes – different locations on the grain boundary may have different grain boundary planes.

7) Grain boundary segregation is directly influenced by the characters of the grain boundary, e.g., specific grain boundary plane, misorientation angle, and grain boundary nature (CSL or random boundary). The authors did not consider this. The planar fluctuation in grain boundary segregation may be influenced by the different characters of different grain boundary planes, even just considering one particular grain boundary.

We greatly appreciate the suggestions of the reviewers and we thoroughly revised our manuscript (NCOMMS-17-22286-T), entitled

“Planar spinodal fluctuation on grain boundaries: a new approach to interface manipulation”

submitted before to *Nature Communications* based on reviewer’s comments. New experimental and theoretical investigations were added (four figures in total). The new title of the manuscript is:

“Confined spinodal fluctuations at crystal defects: a pathway for multistep phase nucleation”.

The answers to the reviewer’s comments are as follows.

Reviewer #1

Opening comment: “This paper reports the discovery of a spinodal fluctuation confined to the planar interface (i.e. within the grain boundary region) in a Fe-Mn alloy. This is, to the best of my knowledge, the first such observation of this type of a grain boundary feature and it represents a significant advance to the field and therefore is fully appropriate for consideration for publication in Nature Communications. The experimental observations obtained from atom probe tomography are well supported by complementary theoretical calculations and simulations. I have a few comments for the authors to consider: (...)”

Answer: We would like to cordially thank the reviewer for the strong support and important comments. We also highly appreciate that the reviewer so clearly emphasizes that this is the **‘first such observation of this type of a grain boundary feature and it represents a significant advance to the field’**.

Comment 1: “The opening statement on line 28 that asserts that spinodal decomposition is the most important phenomena in multi-phase materials is an overstatement in my opinion. There are lots of other phenomena that are arguably equally or more important so I would advise toning down that remark.”

Answer: We fully agree with the reviewer that the assertion that “spinodal decomposition is the most important phenomena in multi-phase materials” would be an overstatement. Nevertheless, the sentence asserts that “spinodal decomposition is one of the most important phenomena in multi-component solids and fluids”, what seems a reasonable statement.

Comment 2: “Line 44 refers to the grain boundary as “interfaces” – plural, when it should be singular “interface” since, as the authors know a grain boundary is one interface thermodynamically.”

Answer: Thank you for the pertinent suggestion. We fully agree. The manuscript has been anyway modified in order to include now results from both low angle and high angle grain boundaries and dislocations.

Comment 3: “Line 62 refers to the grain boundaries as “high angle grain boundaries” – can they be more specific (range of angles, sigma values?).”

Answer: Very good point. Yes, we included now more detailed information about the misorientation of the grain boundaries and how we calculated it. The calculated misorientation values are not typical of any specific sigma boundary. Also, CSL boundaries are rare in BCC iron.

Comment 4: “A key point of the paper is that the spinodal fluctuation is confined to a planar region inside a grain boundary – can they state more precisely how the grain boundary width is defined to further validate that statement – both structurally and chemically.”

Answer: Very good question. The interface composition plots were computed using a method developed in-house in MATLAB (Mathworks inc.). First, a DBSCAN algorithm or a nearest neighbor (NN) filter was used to select the solute atoms which are segregated along the interface. According to their local density maximum (center of mass COM), the interface plane was defined. By splitting the dataset using a Voronoi tessellation, local cumulative curves can be derived and therefrom, the segregation width and compositional values can be extracted, as schematically presented in the figure below. In this sense, the grain boundary region has here been defined as the region enriched with solutes. It must be considered in that context also that APT does in the current case and for many other solid solution metallic systems not have a high enough structural resolution to enable quantification of the solute arrangement along all of the different crystallographic directions or segment normal. Nevertheless, it seems very unlikely that the observed fluctuations could be attributed to different grain boundary planes, since they occur every few nanometers and along the dislocations as well.

Comment 5: “The authors seemed to have intentionally avoided the use of the term “complexion”, which is somewhat puzzling since it appears to fit perfectly with the definition of a grain boundary complexion as an equilibrium state of a grain boundary. Using the term complexion would emphasize the phase-like nature of this grain boundary spinodal. I note that this group did adopt the term complexion in a recent Science paper as applied to dislocations, so I think by not applying it

consistently here it may give the wrong impression that the structure is not an equilibrium feature of the grain boundary, diminishing the novelty of the observation. I would recommend going one step further and including the term complexion in the title. This is a neat discovery of a new type of grain boundary complexion worth highlighting.”

Answer: This is an excellent point which we also discussed in our group extensively in the current context. Thus, we would like to thank the reviewer for this important comment. We did by purpose chose to not adopt the term “complexion” in this specific context as it did not seem entirely appropriate to describe the phenomena we observe here, since spinodal fluctuations are transient states that act as a pathway for phase transition and not as a specific equilibrium state of the boundary. This seems to be a different situation compared to observations on more stable (i.e. less transient) interface complexion states. In order to actually check this we performed additional APT experiments in Fe9Mn samples that were annealed for 2 months at 450°C and we did not observe any dislocation or grain boundary that is enriched with Mn in such a spinodal type transient way, but we instead observed a high volume fraction of austenite that had formed on LAGB and HAGB. These results suggest that the spinodal segregation feature that we found here is not by itself part of the minimum energy state of the system as it is in the case of complexions. Instead the phenomenon we observed here seems to be indeed of a transient nature. We do not think this diminishes the novelty of the observation, since now we can better understand the role of segregation and a confined spinodal as a precursor phenomenon for the nucleation of a new solid state phase. Naturally, the segregation behavior could potentially be described by assuming a steady-state local equilibrium between the matrix and the grain boundary, but this does not imply that the segregation is a stable state. In order to pick up this idea though we could indeed think of using a term that reflects the transient spinodal type feature and reconciles it with the complexion concept, coining it for instance by the term “metastable complexion” to emphasize that we deal here with a decomposition phenomenon that occurs and is to some extent stabilized by the presence of a grain boundary. We also discussed now the concept of complexions in the introduction in more detailed manner regarding the current observation, also reflecting better the comments rendered here in this letter.

Comment 6: “I am not an expert in the quantification of atom probe tomography data but I do know that reconstruction of atom positions is problematic. The changes in Mn concentration are clearly very large and not aberrations of the APT technique, but perhaps some more details about the reconstruction method (other than “we used the IVAS method”) and perhaps a mass spectrum would be useful.”

Answer: We fully agree and provide more details now in the revised version of the paper. The reconstruction was carried out using the protocol introduced by Geiser *et. al* Microscopy and Microanalysis 15, 292-293 (2009) and detailed in Gault et al. Ultramicroscopy 111(6) 448-457. Most of the datasets were reconstructed in Voltage mode (with the exception of the data shown in Fig. 7). The typical detector maps presented in Fig. 2 are strongly indicative of the high reliability of the measurement. The detector reveals elements of the local crystallography of the sample, allowing for atom probe crystallography analysis of the dataset. We could as well calibrate the reconstruction by the interplanar distance of the crystallographic planes associated with the low-hit density poles. These details were included in the manuscript.

Comment 7: “Congratulations on a nice piece of work!”

Answer: We would like to very cordially thank the reviewer for the strong support and for the important comments.

Reviewer #2

Opening comment: “The authors report on a quasi-2D spinodal decomposition that takes place in grain boundaries in Fe-9 at% Mn alloys after a suitable heat treatment. This decomposition is observed and analyzed using 3D atom probe, which allows to map the local Mn composition in grain boundaries. These observations are discussed using thermodynamic data available for this system. These authors conclude that Mn segregation at a grain boundary increases the local Mn composition to the point where this solid solution becomes unstable and undergoes phase separation by spinodal decomposition. Some 2D simulations are presented, supporting the possibility of spinodal decomposition in 2D for this alloy system. The results presented in the manuscript are certainly interesting and worth publishing but in my opinion they do not meet the criteria for publication in Nature Communications for several reasons.”

Answer: We would like to cordially thank the reviewer for the careful analysis of our manuscript and for the helpful comments. We would like to give in sequence a detailed response to these points.

Comment 1: “First, the possibility that segregation at defects could lead to solute enrichment large enough to trigger spinodal decomposition is not a new finding. This pathway has been studied both experimentally and by computer simulations in the case where the defect is a free surface, see for instance C. Helms, Surface Science 69 (1977) 689; Y. Liu and P. Wynblatt, Surface Science 310 (1994) 27; C. Geng, Surface Science 355 (1996) 229. Furthermore the analysis provided in the paper is very limited and omits some key points, regarding for instance the role of in-plane versus out-of-plane diffusivity, whether the thermodynamic functions would correctly predict decomposition in 3D, and whether it is legitimate to use the same expressions for assessing the 2D stability of concentrated solutions in a grain boundary.”

Answer: Many thanks for this hint. Regarding the concerns about the novelty of the work, we do surely recognize that such phenomena have been already studied for surfaces, and we mentioned this also explicitly and clearly in our manuscript. Yet, we did nowhere in our paper claim to observe spinodal decomposition at a surface but instead at grain boundaries. As also the two other reviewers have clearly stated and confirmed, this is indeed the first time that this effect has been experimentally observed. The constraints and boundary conditions for such a phenomenon at a grain boundary are quite different from those encountered at a free surface and we therefore think that the paper actually reports a truly novel effect such as also emphasized by the other two reviewers. Also it should be considered that grain boundaries occur everywhere in materials and have very high relevance for their properties so that understanding such phenomena is of high relevance to any polycrystalline material.

This effect has never been characterized at such fine scales with near atomic resolution. Also, grain boundaries are among the most ubiquitous and least understood types of defects so that such effects may play a fundamental role for any polycrystalline material. The paper makes no claim of any kind

of such phenomena at surfaces. Grain boundaries are also fundamentally different from surfaces and determine very different material features. As mentioned in the article, the solid-state nucleation of a new phase remains poorly understood and the discovery we report here represents an important contribution to the field as the other two reviewers also appreciate.

Comment 2: “Second, the phase field modeling does not provide any new information. The authors would have to justify their choice for the atomic mobility, the interface gradient energy and other parameters if they were trying to extract from the simulations more than just a qualitative agreement, which was expected even before running the simulations.”

Answer: The phase field modeling was introduced for two reasons: First, to show qualitative agreement with the experiments thus supporting the trend we observe experimentally and second, to evaluate whether thermodynamic functions would correctly predict spinodal decomposition. We introduced a short discussion about how we calculated the coherency strain energy and the gradient energy coefficient. We used the intrinsic diffusion coefficient of Mn in BCC iron ($3.0 \times 10^{-18} \text{ nm}^2/\text{s}$) at 450°C to estimate the mobility. We fully agree that it is important to acknowledge that the modelling does not represent the very same situation as the segregation induced spinodal fluctuation observed experimentally, nevertheless it consistently adds important information on the system and its overall thermodynamic trend.

Comment 3: “The experimental findings themselves appear to be solid, the data well analyzed, but somewhat limited. For instance, it would have been useful to have some structural characterization of the grain boundaries, for instance by TEM, to determine whether the regions enriched in Mn correspond to grain boundary dislocation intersections or other specific features of the GBs. In addition, it would have been important to image grain boundaries at an early stage, to see whether segregation first produces a uniformly enriched Mn layer, before reaching a composition that triggered 2D decomposition. This information, it seems, would be essential to complement the present findings.”

Answer: We would like to thank the reviewer for this comment. We fully comply with the reviewer on this point. New experiments (Transmission Kikuchi Diffraction (TKD) and EBSD) as well as further APT analysis (i.e. crystallographic APT analysis and experiments from longer annealing times) were also added to the new version of the paper in order to supply the reader with more detailed information about the nature of the grain boundaries. We have not added further experiments from shorter annealing times because this has been already the object of previous publications (e.g., Kuzmina, M., Ponge, D. & Raabe, D. *Acta Materialia* 86, 182-192 (2015)). It is important to emphasize, however, that we do not expect necessarily a uniformly enriched Mn layer at lower annealing times, since at lower concentrations the GB composition is more affected by structural inhomogeneity and the spinodal fluctuations evolve during the segregation.

Comment 4: “Lastly, even if all the analysis given in the manuscript is correct, it seems that this particular reaction would not be common, as the conditions required for it to proceed seem quite restrictive. The authors for instance did not discuss when this particular kinetic pathway evolved here

would be overridden by the direct nucleation of precipitates in the grain boundaries, or by interface-directed spinodal decomposition (by analogy to surface-directed spinodal decomposition).”

Answer: We would like to thank the reviewer for this comment, but we do not see why the reviewer asserts that this particular reaction would not be common or why the conditions required for it to proceed would be restrictive. We can mention that most of the thermodynamic systems with positive enthalpy of mixing in the solid state and a miscibility gap at lower temperatures are expected to follow a similar transformation pathway even when the global composition is outside the region of metastability. Examples include some of the most important metallic alloy systems for industrial, infrastructure and mobility applications such as steels, aluminium and magnesium alloys. Additionally, many systems might present a “hidden” spinodal region at low temperatures that has been neglected due to our limitations to assess thermodynamic properties at low temperatures. The Fe-C and Fe-N system are well known examples of such a case (see, for example, R. Naraghi, M. Selleby and J. Ågren. *Calphad*. V 46, pp. 148-158 (2014)). We are currently working on a new paper detailing the theoretical aspects of the topic which does not directly fall in the scope of the present article and was hence not included in the current manuscript due to the limited space.

Reviewer #3

Opening comment: “Planar spinodal fluctuation at grain boundaries were studied in this manuscript. Planar spinodal fluctuation is an interesting and novel phenomenon or mechanism for grain boundary segregation. However, the manuscript is not well written, and there are many problems with it, which are detailed as follows. Consequently, the reviewer recommends rejection of the manuscript.”

Answer: We would like to cordially thank the reviewer for the suggestions which certainly help us to improve the quality of the manuscript.

Comment 1: “The English of the manuscript needs to be improved. There are many grammatical errors and inaccurate or confusing phrases/expressions in the manuscript.”

Answer: We thank the reviewer for this hint. The manuscript was very carefully rewritten in order to avoid grammatical errors and inaccurate or confusing phrases/expressions in the manuscript.

Comment 2: “The title includes the wording of “a new approach to interface manipulation”, which is inappropriate. Again, an interesting phenomenon or mechanism is found for grain boundary segregation by the authors. However, this is not really “interface manipulation”. Interface manipulation was not really discussed in the manuscript.”

Answer: The manuscript has a new title now: “Confined spinodal fluctuations at crystal defects: a pathway for multistep phase nucleation”. New results were incorporated in this new version in order to support the findings and justify the title.

Comment 3: “The abstract is not well written. The first couple of sentences are not really relevant to the study describing in the manuscript (e.g., mechanical properties or strengthening mechanisms). It is claimed that a new phase transition mechanism is discovered, however, the manuscript or the study is really on grain boundary segregation, and phase transition was barely discussed in the manuscript.”

Answer: We fully comply. The first couple of sentences related to mechanical properties and strengthening mechanisms were removed from the abstract. More experimental details were added in order to support the claim that the spinodal fluctuations can act as pathway to phase transition.

Comment 4: “The introduction part of the manuscript is not well written. Essentially, there is only one paragraph for the introduction section, only describing spinodal decomposition. Background and motivation for this study should have been clearly elucidated in the introduction section. For example, what is the current state-of-art on studies on grain boundary segregation, why is this important, and why did the authors choose Fe-Mn system.”

Answer: We would like to thank the reviewer for this comment. The previous manuscript was written according to the constraints associated to the submission of Letters to Nature, and we followed the advice from the editor of the sister journal and transferred the manuscript to Nature Communications. We have now included two more paragraphs in order to elucidate the importance of segregation and spinodal decomposition on the nucleation of a new phase. We did not upfront add details about the Fe-Mn system specifically because the paper is not focused on properties/applications, but rather on a new mechanism for phase nucleation.

Comment 5: “In the manuscript, adsorption and grain boundary segregation are used interchangeably. Although they have some similarity, adsorption and grain boundary segregation are different. Adsorption describes the adhesion of atoms from a gas or liquid phase onto a surface, whereas grain boundary segregation describes the segregation of atoms in a solid to a grain boundary.”

Answer: We would like to thank the reviewer for this comment, but we have to disagree on this point. Segregation and adsorption are essentially synonymous from the thermodynamics point of view. In practice, adsorption is more often used to describe the partitioning of solutes between the free surface and a gas or a liquid, while segregation is used to describe the partitioning of solutes between an internal interface (like grain boundaries) and the adjoining solid solution, but they are often used interchangeably as well (e.g., Gibbs adsorption/segregation isotherm). We followed here the same wording used in numerous textbooks or review papers about adsorption, segregation, wetting and related phenomena (e.g., Lejcek, P. in Grain Boundary Segregation in Metals Vol. 136 Springer Series in Materials Science 1-239 (2010)).

Comment 6: “A grain boundary may not be just one plane. In the manuscript, the authors assumed that a grain boundary is just one flat plane. However, a grain boundary may be curved, and it may have multiple planes – different locations on the grain boundary may have different grain boundary planes.”

Answer: We did not assume or suggest anywhere in the paper that a grain boundary is just one flat plane. This misunderstanding possibly arises from the use of the expression “in-plane concentration”. Certainly the boundary is curved, this is why we introduced the method to investigate the local composition and added the following sentence: “First, a DBSCAN algorithm or a nearest neighbor (NN) filter was used to select the solute atoms which are segregated along the interface. According to their local density maximum (center of mass COM), the interface plane was defined. By splitting the dataset using a Voronoi tessellation, local cumulative curves can be derived⁵⁴ and therefrom, the

segregation width and compositional values can be extracted". The composition was computed from the local normal to the grain boundary surface, although we project it in 2D map for better visualization.

Comment 7: "Grain boundary segregation is directly influenced by the characters of the grain boundary, e.g., specific grain boundary plane, misorientation angle, and grain boundary nature (CSL or random boundary). The authors did not consider this. The planar fluctuation in grain boundary segregation may be influenced by the different characters of different grain boundary planes, even just considering one particular grain boundary."

Answer: The reviewer is correct in asserting that the segregation behavior is influenced by the specific grain boundary plane, misorientation angle, and grain boundary character. We did not neglect this fact, but it is very unlikely that the observed composition fluctuations can be explained only on this base. We added new experimental results showing that dislocations display the very same type of compositional fluctuation: a base level around 15 at% Mn together with fluctuations ranging between 15at% Mn and 25-30 at% Mn. Further, the constituent dislocations of the LAGB shows a very similar pattern. It is true that HAGB have higher excess of solute than LAGB when normalized by area, but the individual dislocations in a LAGB shows a compositional variation similar to a HAGB. Also, CSL boundaries are rare in BCC iron. It is very unlikely that this high periodical composition variation observed both in dislocations, HAGB and LAGB can be attributed to different characters of different grain boundary planes.

REVIEWERS' COMMENTS:

Reviewer #4 (Remarks to the Author):

This is a nice piece of work that deserves to be published.

As regards GB segregation, I would suggest the authors to cite the pioneer work of Mc Lean.

The authors should also stress on the fact that periodic concentration fluctuations along GBs do not necessary originate from instabilities, i.E. 2D spinodal decomposition. I suggest the authors to cite the APT work of NWU group (D.N. Seidman et al.) or that of GPM at the university of Rouen on nickel base superalloys who showed, preferential segregation of boron on preferential sites of GBs that are periodically spaced (e.g. D. Lemrhand et al. Phil. Mag. (2002)).

We greatly appreciate the suggestions of the reviewer on our manuscript NCOMMS-17-22286A-Z. We have now revised it based on the reviewer's comments.

The answers to the reviewer's comments are as follows.

Reviewer #4

Opening comment: "This is a nice piece of work that deserves to be published."

Answer: We would like to cordially thank the reviewer for the strong support and important comment.

Comment 1: "As regards GB segregation, I would suggest the authors to cite the pioneer work of Mc Lean."

Answer: We thank the reviewer for the pertinent suggestion. Mc Lean's work is now cited in the paper.

Comment 2: "The authors should also stress on the fact that periodic concentration fluctuations along GBs do not necessary originate from instabilities, i.E. 2D spinodal decomposition. I suggest the authors to cite the APT work of NWU group (D.N. Seidman et al.) or that of GPM at the university of Rouen on nickel base superalloys who showed, preferential segregation of boron on preferential sites of GBs that are periodically spaced (e.g. D. Lemarchand et al. Phil. Mag. (2002))."

Answer: Again, we thank the reviewer for this suggestion. We added the following sentence and quoted the suggested reference as well as others: "The linear and planar compositional fluctuations observed here are visually very similar to those reported for bulk spinodal decomposition and contrast with other studies that show preferential segregation of solutes to periodically spaced sites at grain boundaries".

We kindly thank you for the time spent processing our manuscript and hope that it is now suitable for publication in **Nature Communications**.